# Developmental Risk, Adversity Experiences and ADHD Clinical Profiles: A Naturalistic Exploratory Study

**DOI:** 10.3390/brainsci12070919

**Published:** 2022-07-13

**Authors:** Brooke Streeter, Joseph Sadek

**Affiliations:** 1Faculty of Sciences (Medical Sciences), Dalhousie University, Halifax, NS B3H 4R2, Canada; br611633@dal.ca; 2Department of Psychiatry, Dalhousie University, Halifax, NS B3H 4R2, Canada

**Keywords:** attention-deficit/hyperactivity disorder, ADHD, risk and adversity-factors, severity, comorbidities, clinical profile

## Abstract

Attention-deficit/hyperactivity disorder (ADHD) is a persistent neurodevelopmental disorder that results from complex interactions of multiple genes and environmental risk and adversity factors. Some researchers have suggested a need for additional research into differing clinical presentations of ADHD for further classification. In this context, this study aimed to investigate whether increases in risk and adversity factors increase the severity of ADHD and the number of comorbid psychiatric disorders. This is a naturalistic retrospective chart review exploratory study in 100 patients 16 years or older who have a confirmed diagnosis of ADHD. The quantitative data were analyzed using SPSS, using the Mann–Whitney test for parametric data and the Chi-square and Kruskal–Wallis *p* value tests for non-parametric and categorical data. Qualitative data were tabulated and described. The study found that (1) the average number of comorbidities increases with the severity of ADHD, (2) the average number of risk and adversity factors increases with the severity of ADHD, (3) the number of risk and adversity factors were positively associated with the number of comorbidities, and (4) Level of education was negatively associated with the number of risk and adversity factors and the number of comorbidities. The implications of these findings are discussed, and future research in this important area is suggested.

## 1. Background

Attention-deficit/hyperactivity disorder (ADHD) is a persistent neurodevelopmental disorder that affects around 6% of children and adolescents and 3% of adults worldwide. ADHD can present as challenges in sustained attention, motor hyperactivity, and enhanced impulsivity. In addition, throughout an individual’s lifetime, ADHD can increase the risk and adversity of other psychiatric disorders, educational and occupational failure, accidents and injuries, criminality, social difficulties, and substance use disorders. ADHD does not have one direct cause; instead, it results from complex interactions of multiple genes and environmental risk and adversity factors [1].

While it is generally accepted that the disorder is highly heritable (>70%), it is estimated that between 10 and 40% of the variance associated with ADHD is likely to be accounted for by environmental factors [2].

ADHD has been associated with various risk and adversity factors, but most of these associations have yet to be shown as causal. Prematurity—especially extreme prematurity—maternal smoking or substance use during pregnancy, low Apgar scores at 5 min, low birth weight, and other labor/delivery complications are associated with ADHD [2].

Several general psychosocial adversity factors have been studied concerning ADHD [2,3]. Among the factors associated with ADHD are low parental education, low income and poverty, deprivation, hostile parenting, bullying, adverse parenting practices, peer victimization, and family discord [3,4,5,6,7,8,9,10,11,12,13,14,15,16,17,18,19,20]. A meta-analysis of the dimensions of SES and their association with ADHD indicate that children in families of low socioeconomic status (SES) are, on average, 1.85–2.21 times more likely to have ADHD [21]. ADHD-related difficulties with higher-order cognitive processes affect encoding and utilization of expectations regarding the effects of marijuana [22]. Compared with children without ADHD, children with ADHD have higher Adverse Childhood Experience exposure. In addition, a significant correlation has been observed between adverse childhood experiences and moderate to severe levels of ADHD [23].

Comorbidities are characterized by more than one specific condition in an individual and are common among psychiatric disorders. Approximately 75% of persons with ADHD have an additional mental disorder, and around 60% have multiple comorbid mental disorders [24]. High rates of comorbidities with ADHD have been reported in both clinical samples and epidemiological studies [25]. Authors have indicated that comorbidity is generally high in ADHD in both children and adults. Comorbid disorders may share common vulnerability, genetic, or psychosocial adversity factors. They may present as a separate entity or as an expression of phenotypic variability [26,27].

Researchers continue to attempt to find additional subtypes of ADHD, such as Sluggish Cognitive Tempo described by Barkley [28]. Some researchers indicate a need for further research to investigate the association of the risk and adversity factors, comorbidities, and clinical profiles in ADHD [29]—notably since earlier studies suggested that certain risk and adversity factors are associated with a specific subtype of ADHD in genetically susceptible individuals [30]. ADHD is considered a neurodevelopmental disorder. The neurodevelopmental theory holds that a disruption of normal brain development in utero or during early life underlies the subsequent development of symptoms later in life [31]. Recent research on a neurovisceral integration model (NVI) has proposed that physiological, emotional, and cognitive regulation processes are related to each other in the control of behavior and adaptability [32]. The results of functional imaging studies in ADHD indicate multiple abnormalities not limited to the frontal–striatal circuitry—which is crucial for executive/motivational function—but also includes the parietal, temporal, and motor cortex, as well as the cerebellum [33,34]. The ventral striatum, including the caudate nucleus, nucleus accumbens, and the putamen, shows lower activation during reward anticipation in ADHD than in controls—this may be related to hyperactive–impulsive symptom severity rather than inattention [35,36,37]. Questions are raised about the role of additional metabolic pathways such as the tryptophan (TRP)–Kynurenine (KYN) pathway in different psychiatric disorders [38]. The brain’s executive system is hypothesized to control inhibition, working memory, and cognitive flexibility—particularly when more complex demands require adaptation and effortful control [39,40,41]. The complexity of the pathophysiology of ADHD as a neurodevelopmental disorder creates additional importance for research on risk factors and comorbidities.

## 2. Objective

This exploratory study aimed to examine the association of different environmental risk factors, adverse childhood experiences, and comorbid psychiatric disorders associated with ADHD, as well as to understand whether certain etiological risk factors are associated with a specific clinical profile of ADHD.

We hypothesized that the severity and number of adverse risk factors in childhood that individuals have experienced are correlated with the severity of ADHD symptoms.

## 3. Methods

### 3.1. Study Design

This is a naturalistic retrospective chart review pilot study that uses existing data collected through patient charts. The patient charts include a psychiatric report, intake forms, treatment records, and various questionnaires that include numerous developmental risk and adversity factors. This data was obtained through the electronic medical records of the Med access account of the outpatient psychiatric clinic specializing in ADHD where the study occurred. Patients of the clinic signed a consent form upon their first visit to the clinic allowing the use of their anonymized health information in research. To ensure the complete privacy and confidentiality of all the participants, the names attached to the data were erased. Each patient was assigned a participant number to conserve their anonymity moving forward.

### 3.2. Inclusion and Exclusion Criteria

The study included all patients 16 years or older with a confirmed diagnosis of ADHD by a psychiatrist from January 2021 until September 2021, reaching a maximum of 100 participants. From the collected data, the presence of developmental risk and adversity factors, symptom severity, and comorbidities was investigated. Exclusion criteria included the lack of a confirmed ADHD diagnosis and evidence of malingering as confirmed by continuous performance test results (CPT).

### 3.3. Data Collection and Analysis

Data collection included demographic information on gender and age, ADHD diagnosis, the severity of ADHD symptoms based on DSM-V criteria (mild, moderate, and severe) [22], and different psychiatric comorbidities. All patients had a full psychiatric assessment and ADHD assessment by a psychiatrist. The clinic also required all patients coming in with concerns of ADHD to complete a computerized continuous performance test (CPT). CPT scores were collected. Other information collected included suicide risk and adversity level, drug and alcohol history and current use, medical history, psychiatric history, family history, and personal and social history. Adversity factors collected included prenatal and perinatal risk and adversity factors, including maternal alcohol consumption, smoking, maternal stress, birth complications, low birth weight, and prematurity. Other adverse childhood experiences that were considered included sexual or physical abuse, bullying, academic difficulties, being disliked by teachers, parental separation, and socioeconomic adversity factors while growing up.

The comorbid disorders this study assessed were Alcohol and Substance Use Disorder, Smoking, Binge Eating Disorder, Bipolar and Unipolar mood disorders, Generalized Anxiety Disorder (GAD), Phobia, Social Anxiety Disorder, Post-traumatic Stress Disorder (PTSD), tics, Obsessive–compulsive Disorder (OCD), Schizophrenia and related disorders, Specific Learning Disorder, Anti-social Personality Disorder (ASPD), Borderline Personality Disorder (BPD), history of Oppositional Defiant Disorder, and Conduct disorder.

For ADHD symptoms, we looked at the number and severity of symptoms (mild, moderate, or severe) and subtype (inattentive, hyperactive–impulsive, or combined type).

### 3.4. Statistical Analysis

A qualitative analysis was completed for all relevant criteria where we identified themes in participants’ history. Descriptive statistics (mean and standard deviation) were used to analyze the data.

The quantitative data were analyzed using SPSS (Version 28.0., IBM Corp, Armonk, NY, USA), using the Mann–Whitney test for parametric data and the Chi-square and Kruskal–Wallis *p* value tests for non-parametric and categorical data. Qualitative data were tabulated and described.

## 4. Results

The age-group range of the sample was 17 to 61, with a mean age of 29. The sample was composed of 30 males and 70 females; 51% of the patients had a college education or higher; 49% of the patients had inattentive type ADHD. Generalized anxiety disorder was a comorbidity in 93% of the patients. Table 1 describes the characteristics of the study sample.

The patients reported growing up in poverty or with low SES (40%), using Marijuana more than three days per week (42%), having academic difficulties (59%), and having a history of self-harm (63%; Table 2).

The study found that (1) the average number of comorbidities increases with the severity of ADHD (Figure 1), (2) the average number of risk and adversity factors increases with the severity of ADHD, (3) the number of risk and adversity factors is positively correlated with the number of comorbidities; R = 0.85; Figure 2), and (4) Level of education is negatively correlated with the number of risk and adversity factors (R = −0.89) and the number of comorbidities (R = −0.58) where R is the correlation coefficient.

The statistical analysis showed that the ADHD Combined type may be more likely to be associated with more risk and adversity factors than the Inattentive type. In addition, Inattentive and Hyperactive/Impulsive types were more likely to be associated with mild or moderate severity, whereas the Combined type was more commonly associated with severe symptoms.

There are three categories of ADHD severity: mild, moderate, and severe. Participants with mild ADHD exhibited an average number of comorbidities of 1.89 and an average number of risk and adversity factors of 4.78; moderate ADHD had an average number of comorbidities of 2.30 and an average number of risk and adversity factors of 6.12; severe ADHD had an average number of comorbidities of 2.25 and an average number of risk and adversity factors of 7.15. The overall number of risk and adversity factors increased with the increased severity of ADHD and comorbidities.

This study sorted levels of education into five groups: Less than Grade 12, Grade 12, college, university and postgraduate. If participants had an education of less than grade 12, their average number of risk and adversity factors was 9.6; if they had completed Grade 12, then their average number of risk and adversity factors was 6.7; if they had completed college, their average number of risk and adversity factors was 6.4; if they had completed university, their average number of risk and adversity factors was 5.4; and if they had completed postgraduate studies, their average number of risk and adversity factors was also 5.4 (Figure 3).

For the statistical analysis, we used the Chi-Square *p* value. The risk and adversity factor prevalence were divided into two categories: six or fewer risk and adversity factors (*N* = 60) and seven or more risk and adversity factors (*N* = 40). When looking at the three subtypes of ADHD and the number of adversity factors (Table 3), the majority of patients with the Inattentive subtype (58%, *N* = 35) had six or fewer adversity factors. The majority of patients with the Combined subtype (55% *N* = 43) had seven or more adversity factors. The hyperactive–impulsive type exhibited the fewest adversity factors out of the total population (*N* = 9) and did not show much significance. Only 32% of patients with the inattentive subtype had seven or more risk and adversity factors. Only 35% of patients with the Combined subtype (*N* = 21) had six or fewer risk and adversity factors, with an overall *p* value of 0.0392.

Considering the subsequent risk and adversity factors, 13 of the 19 proposed risk and adversity factors showed statistical significance. Of those with a history of self-harm, 32 (53.3%) had six or fewer risk and adversity factors and 31 (77.5%) had seven or more risk and adversity factors (*p* = 0.0142). Of those who smoked, 13 (21.7%) had six or fewer risk and adversity factors and 20 (50.0%) had seven or more risk and adversity factors (*p* = 0.0032). Of those who used marijuana more than three days per week (50%), 20 (33.3%) had six or fewer risk and adversity factors and 22 (55.0%) had seven or more risk and adversity factors (*p* = 0.0315). Of those who drank more than three days per week (50%), two (3.3%) had six or less risk and adversity factors and seven (17.5%) had seven or more risk and adversity factors (*p* = 0.0153). Of those who used other drugs, 8 (13.3%) had six or fewer risk and adversity factors and 15 (37.5%) had seven or more risk and adversity factors, (*p* = 0.0049). Of those whose parents split before the age of 10. 18 (30.0%) had six or fewer risk and adversity factors and 21 (52.5%) had seven or more risk and adversity factors (*p* = 0.0238). Of those who had complications at birth, 13 (21.7%) had six or fewer risk and adversity factors and 18 (45.0%) had seven or more risk and adversity factors (*p* = 0.0135). Of those who experienced academic difficulties, 29 (48.3%) had six or fewer risk and adversity factors and 30 (75.0%) had seven or more risk and adversity factors (*p* = 0.0079). Of those who had failed grades, 9 (15.0%) had six or fewer risk and adversity factors and 14 (35.0%) had seven or more risk and adversity factors (*p* = 0.0199). Of those who were not liked by teachers, 12 (20.0%) had six or fewer risk and adversity factors and 17 (42.5%) had seven or more risk and adversity factors (*p* = 0.0151). Of those who had a poor current relationship with parents, 6 (10.0%) had six or fewer risk and adversity factors and 21 (52.5%) had seven or more risk and adversity factors (*p* < 0.0001). Of those who had been charged by police, two (3.3%) had six or fewer risk and adversity factors and eight (20.0%) had seven or more risk and adversity factors, (*p* = 0.0065). Of those who had been arrested, two (3.3%) had six or fewer risk and adversity factors and eight (20.0%) had seven or more risk and adversity factors (*p* = 0.0065; Table 4).

For the analysis of severity and ADHD subtype, the severity of symptoms was divided into two categories: Mild/Moderate (*N* = 52) and Severe (*N* = 48). When looking at the three subtypes of ADHD, if Mild/Moderate Severity (*N* = 52), 13 (25%) exhibited the Combined type, 6 (11.5%) exhibited the Hyperactive/Impulsive type, and 33 (63.5%) exhibited the Inattentive type; whereas, if Severe (*N* = 48), 30 (62.5%) exhibited the Combined type, 3 (6.3%) exhibited the Hyperactive/Impulsive type and 15 (31.3%) exhibited the Inattentive type. Overall, these findings exhibited a *p* value of 0.0008.

When comparing the severity of ADHD to the ADHD subtype, the Inattentive type was most likely to exhibit Mild/Moderate severity (63.5%). The Combined type was more likely to exhibit severe ADHD (62.5%). The hyperactive–impulsive type was exhibited the least out of the total population (*N* = 9) and did not show much significance. However, all 13 of the listed adversity factors indicated in Table 5 were more likely to be exhibited by a participant with seven or more adversity factors than six or fewer adversity factors.

The risk and adversity factors of smoking and not being liked by teachers were the only two risk and adversity factors significantly associated with the severity of symptoms. For smoking, if Mild/Moderate severity (*N* = 52), 9 (17.3%) were smokers—whereas if Severe (*N* = 48), 24 (50%) were smokers, with a *p* value of 0.0005. For not being liked by teachers if Mild, Moderate Severity (*N* = 52), 8 (15.4%) were not liked by teachers—whereas if Severe (*N* = 48), 21 (43.8%) were not liked by teachers, with a *p* value of 0.0018 (Table 6).

## 5. Discussion

This study was beneficial in investigating not only the developmental predisposing risk and adversity factors for ADHD such as birth complications, but also other perpetuating adversity factors such as substance use and self-harm. The study was designed to have a naturalistic sample from a specialized ADHD clinic and linked the risk and adversity factors with comorbid disorders and the severity of the ADHD presentation.

Risk and adversity factors such as delayed developmental milestones may be genetic, environmental, or combined, so separating genetic from environmental risks and adversity was not evident in this study; certainly, the importance of genetic risk and adversity factors on the development of ADHD has been highlighted in several previous studies in the literature [27,36].

Our hypothesis for this study was that the higher the number of risk and adversity factors, the more severe the clinical profile in ADHD, as manifested by the number of ADHD symptoms and the number of comorbid disorders. As per the evidence indicated in Table 3, Table 4 and Table 5, this hypothesis is supported.

The data is consistent with previous studies regarding several ADHD risk and adversity factors and comorbidities. In this study, bullying was our most prevalent adversity factor, with 81% of our participants having experienced it. It is suggested that greater emotional reactivity, social skill difficulties, and impulsive behaviors associated with individuals with ADHD can elicit negative reactions from peers and place them at greater risk of adversity, rejection, and victimization. Several studies have supported this finding [20].

Socioeconomic disadvantage (40%) and Marijuana use (42%) were also some of our most prevalent adversity factors found; their association with ADHD had been previously noted in research carried out by Russell and colleagues (2016) [21] and Harty and colleagues (2015) [22], respectively. Children in families of low socioeconomic status (SES) are on average 1.85–2.21 more likely to have ADHD than their peers in high SES families. The meta-analysis showed evidence of this association despite the between-study heterogeneity [21]. Comorbid generalized anxiety is common with ADHD, and in this sample the high prevalence may reflect a peak in anxiety related to the COVID pandemic in 2021.

The high number of patients who reported self-harm may reflect different comorbidities such as mood disorders, borderline personality disorder, and impulsive behavior.

One interesting result was the negative relationship between the level of education and the number of risk and adversity factors. Education level was collected as part of the demographic information; higher numbers of risk factors were related to lower levels of education.

ADHD has a distinct impact on academic adversity beyond the effects related to the influence of personal (e.g., sociodemographic, personality, prior achievement, specific learning disorder, motivation) and contextual (e.g., school issues) factors. ADHD may compound difficulties with academic outcomes through its effect on cognitive and executive functioning, self-regulation, or other ADHD-related behavior such as poor task completion and off-task behavior [32]. Although poor academic functioning is well documented in ADHD research, our specific finding may have importance in future strategies to improve educational outcomes in patients with ADHD. Despite the trend of higher numbers of risk and adversity factors being associated with lower levels of education that was evident in our raw data, a larger sample size will be required to confirm the statistical significance of these interesting findings.

Some adversity factors such as not being liked by teachers were related to severe ADHD presentation. This interesting finding should be investigated further as teachers may have important effects on different stages of development and the formation of healthy self-esteem.

The growing awareness of mood and anxiety disorders comorbid with ADHD as a neurodevelopmental disorder indicates that the boundaries between neurodevelopmental and non-neurodevelopmental disorders are ambiguous. Some researchers question if the comorbidities are related to the ADHD or to its risk factors [42]. In our study, the majority of patients with the combined subtype had seven or more adversity factors, whereas patients with the predominantly inattentive type had six or fewer risk factors.

Children with predominantly inattentive ADHD are more passive, less aggressive, less assertive, and less knowledgeable of appropriate social behavior than those diagnosed with combined ADHD. They may be socially neglected, versus the combined type who are socially rejected [33].

## 6. Conclusions

This study highlighted the impact of risk and adversity factors in patients with ADHD—particularly that they were associated with symptom severity, comorbidities, and education level. Results showed that the average number of comorbidities increased with the severity of ADHD, as did the average number of risk factors; moreover, the number of risk factors positively correlated with the number of comorbidities, but level of education negatively correlated with the number of risk factors and the number of comorbidities. There may be a distinct profile in patients with ADHD who have severe symptoms, multiple comorbidities, and seven or more risk factors. This specific group may require different management strategies and resources. Designing screening tools for patients with low academic achievements and multiple risk factors might be helpful for detecting comorbidities and distinct ADHD profiles.

## 7. Limitations and Future Directions

This study has several limitations. One of the limitations of this study is the retrospective nature of the chart review; recall bias can be a limiting factor. Since participants fill out the intake form when they come into the clinic for their appointment, it requires them to think back to many experiences they had in the past, such as “being bullied, sexual abuse before age 15, performance in school, etc.” However, the nature and intensity of these adversity factors make recall bias less prevalent. Another limitation is the sample size of the study. Since it is an exploratory study, we have a smaller sample size (100), which limits the analysis of some of the findings. The male to female ratio is opposite to the natural prevalence of ADHD; however, it represents a new, naturalistic trend of more females seeking ADHD assessments. Among the limitations of this study is the lack of an assessment of proactive inhibitory control.

More research with a larger sample is needed in patients with ADHD and multiple comorbidities. Future studies may also wish to investigate the effect of mitigating adversity factors in patients with ADHD, and whether specific management approaches would be more helpful for certain ADHD severities or subtypes. Future research could adapt this study in a larger sample size to allow for more valid findings. There is a clear need for further research in specific areas such as personality disorders, suicide prevention, and ADHD.

## Figures and Tables

**Figure 1 brainsci-12-00919-f001:**
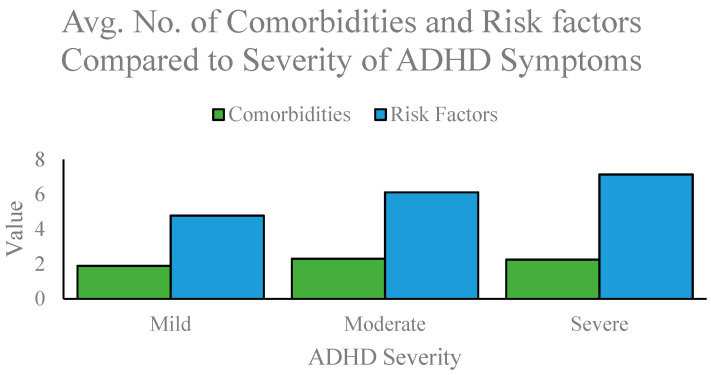
Average number of comorbidities and risk and adversity factors compared to the severity of ADHD Symptoms. Severity of ADHD symptoms measured on the X-axis and mean number of comorbidities and average number of risk and adversity factors measured on the Y-axis.

**Figure 2 brainsci-12-00919-f002:**
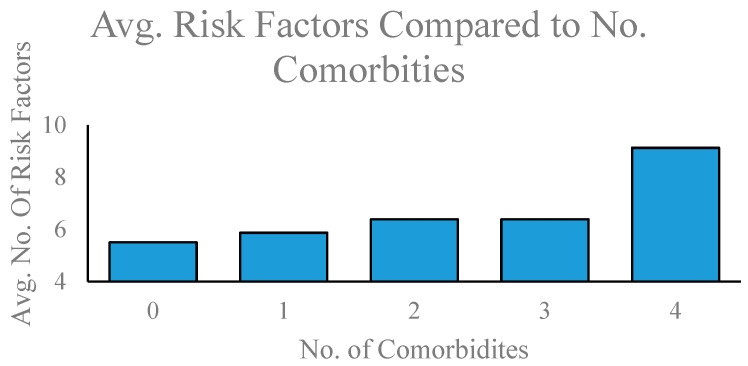
Average number of risk and adversity factors compared to average number of comorbidities. Number of comorbidities measured on the X-axis and mean number of risk and adversity factors measured on the Y-axis. The number of participant comorbidities ranged from 0 to 4, whereas possible risk and adversity factors ranged from 4 to 19.

**Figure 3 brainsci-12-00919-f003:**
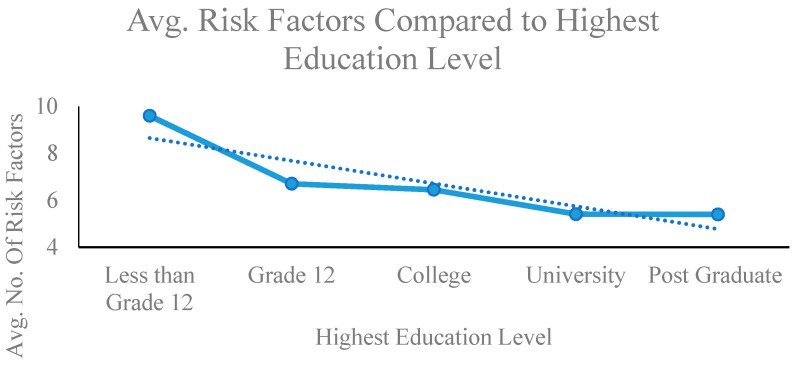
Average risk and adversity factors compared to highest education level. Level of education attained measured on the X-axis and mean number of risk and adversity factors measured on the Y-axis.

**Table 1 brainsci-12-00919-t001:** Characteristics of Study Sample (*N* = 100).

Variables	*N/%*
**Age**	
Range	17–61
Mean	29
**Education**	
Less than grade 12	5
Grade 12	44
College	29
University	17
Post grad	5
**Sex**	
Male	30
Female	70
**ADHD Diagnosis**	
Inattentive Type	48
Hyperactive–Impulsive Type	9
Combined Type	43
**Comorbid disorders**	
Generalized Anxiety Disorder (GAD)	93
Borderline Personality Disorder (BPD)	76
Binge Eating Disorder (BED)	12
Substance use Disorder (SUD)	10
Major Depressive Disorder (MDD)	9
Specific Learning Disorder (SLD)	4
Oppositional Defiant Disorder (ODD)	3
Obsessive Compulsive disorder (OCD)	3
Bipolar Disorder	2
Conduct Disorder	2
Tourette Syndrome (TS)	2
Autism Spectrum Disorder (ASD)	2
Alcohol Use Disorder (AUD)	2
Post-traumatic stress disorder (PTSD)	1
Trichotillomania	1
Social anxiety	1
Specific Phobia	1

**Table 2 brainsci-12-00919-t002:** Most Common Reported Factors Associated with ADHD in this sample.

**Risk and Adversity Factor**	**Total**
Raised in a foster home	0
Drinking more than 3 days per week (50%)	9
Charged by police	10
Arrested	10
Delayed developmental milestones (genetic or environmental)	12
Pregnancy problems	12
Using other drugs	23
Failed Grades	23
Physical abuse	27
Poor current relationship with parents	27
Not liked by teachers	29
Birth complications	31
Smoking	33
Sexual abuse before 15	34
Current suicidal ideation	37
Parents splitting before age of 10	39
Low SES status	40
Using Marijuana more than 3 days per week (50%)	42
Academic Difficulties	59
History of self-harm	63
Bullied as a child	81

**Table 3 brainsci-12-00919-t003:** Association of number of risk and adversity factors as compared to ADHD type.

*ADHD Subtype*	*Six or Fewer Risk and Adversity Factors* (*N* = 60)		*Seven or More Risk and Adversity Factors* (*N* = 40)		*Total*	*p-Value*
	*N*	%	*N*	%	*N*	
Combined Type	21	35	22	55.0	43	0.0392
Hyperactive/Impulsive Type	4	6.7	5	12.5	9	
Inattentive Type	35	58	13	32	48	

**Table 4 brainsci-12-00919-t004:** Association of type of risk and adversity factor as compared to the number of risk and adversity factors.

*Risk and Adversity Factor*	*Six or Fewer Risk and Adversity Factors* (*N* = 60)		*Seven or More Risk and Adversity Factors* (*N* = 40)		*Total*	*p-Value*
	*N*	%	*N*	%	*N*	
History of self-harm	32	53.3	31	77.5	63	0.0142
Smoking	13	21.7	20	50.0	33	0.0032
Using marijuana more than 3 days per week	20	33.3	22	55.0	42	0.0315
Drinking more than 3 days per week	2	3.3	7	17.5	9	0.0153
Using other drugs	8	13.3	15	37.5	23	0.0049
Parents splitting before age of 10	18	30.0	21	52.5	39	0.0238
Birth complications	13	21.7	18	45.0	31	0.0135
Academic difficulties	29	48.3	30	75.0	59	0.0079
Failed Grades	9	15.0	14	35.0	23	0.0199
Not liked by teachers	12	20.0	17	42.5	29	0.0151
Poor current relationship with parents	6	10.0	21	52.5	27	<0.0001
Charged by police	2	3.3	8	20.0	10	0.0065
Arrested	2	3.3	8	20.0	10	0.0065

**Table 5 brainsci-12-00919-t005:** Association between ADHD subtype and Severity of ADHD symptoms.

*ADHD Type*	*Mild, Moderate* *Severity* (*N* = 52)		*Severe* (*N* = 48)		*Total*	*p-Value*
	*N*	%	*N*	%	*N*	
Combined Type	13	25.0	30	62.5	43	0.0008
Hyperactive/Impulsive Type	6	11.5	3	6.3	9	
Inattentive Type	33	63.5	15	31.3	48	

**Table 6 brainsci-12-00919-t006:** Association between certain risk and adversity factors and severity of symptoms.

*Risk and Adversity Factor*	*Mild, Moderate* *Severity* (*N* = 52)		*Severe* (*N* = 48)		*Total*	*p-Value*
	*N*	%	*N*	%	*N*	
Smoking	9	17.3	24	50.0	33	0.0005
Not liked by teachers	8	15.4	21	43.8	29	0.0018

## Data Availability

The data presented in this study are available on request from the corresponding authors.

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
