# Peer review of "Developmental Risk, Adversity Experiences and ADHD Clinical Profiles: A Naturalistic Exploratory Study"

_brainsci, 2022, doi:10.3390/brainsci12070919_

Round 1

Reviewer 1 Report

Comments and Suggestions for Authors

This study explored the characteristics of genetic/environmental risk factors, adverse childhood experiences, and coexisting psychiatric disorders with ADHD and their specific clinical manifestations. Sounds interesting, but I have some questions, as follows:

ABSTRACT

1.The abstract should be succinct, and the methods section will describe the population included and the methods used, and there is no need to list the inclusion and exclusion criteria in detail.

2.The conclusion of ABSTRACT is to clarify what the findings suggest and the theoretical value shown, rather than an outlook for future research.

INTRODUCTION

1.Lines 48-52 have no citations, and the order of references in the manuscript is out of order, and is not cited in ascending order. Author checked and revised.

2.SES does not include its full name on its first appearance. There are also MDD, ODD, SUD, etc. in Table 1. In fact, the authors should check the full text and correct it.

3.Lines 76 to 79, the purpose of the study is to determine the association of different genetic/environ mental risk factors, adverse childhood experiences, and comorbid psychiatric disorders with ADHD and its specific clinical presentation. This purpose appears to have covered lines 80-82 "The study also . . . profile of ADHD." I think the author can adjust the sentence expression and integrate it, otherwise it seems repetitive.

METHOD AND RESULTS

1.It is important that the statistical methods of the article should be listed using a separate subheading.

2.The author describes the results in lines 143 to 145, 220 to 221, and 271 to 272 using a "correlation" statement, how the author's relevant conclusions are obtained, because the author did not use correlation analysis or other methods to analyze? Therefore, in Tables 3, 4, and 5, it is inappropriate to use the title "Correlation of..." because the authors did not do a true correlation analysis. The statement "the increase in the number of the adversity factors increases the severity of ADHD" is acceptable.

3.Figures 1 and 2 are not labeled in the interpretation of the results, although it may be in lines 141 to 145.

4.In Table 2, the author performed a 3-group chi-square test, and the p-value was statistically significant. Did the author conduct further pairwise comparisons to test which groups are different?

5.The authors aim to examine the association of genetic/environmental risk factors, adverse childhood experiences, and comorbid psychiatric disorders with ADHD and its specific clinical presentation. It should be clear which factors are genetic risk factors and which are environmental risk factors. In addition, did the authors exclude the influence of genetic factors when exploring the relationship between environmental risk factors and ADHD?

6.In Table 4, the authors divided the risk factors into ≤6 and ≥7, and then compared each individual risk factor, whether the author excluded the influence of each individual risk factor itself, such as the first History of self In the -harm comparison, did the author exclude the influence of History of self-harm on the results?

DISCUSSION

1.Lines 232 to 250, the author seems to repeat the results again, the discussion is to discuss the results of the manuscript, rather than simply repeating the results.

2.In lines 271 to 277, the author says that a relationship between education level and risk factors has been found. The author's discussion is superficial. Is there any previous literature supporting this result? The author should explain.

CONCLUSION

I don't understand the author's conclusion. It seems that it is all about the prospect and design of future research. The author did not summarize the conclusions of this study? I think this is confusing and the author should rewrite the conclusion section.

Author Response

Dear Reviewer I

Thanks so much for the excellent and very helpful comments. We really appreciate this objective review. We have made all the required adjustments and corrections.

Joseph Sadek

Abstract

.The abstract should be succinct (brief), and the methods section will describe the population included and the methods used, and there is no need to list the inclusion and exclusion criteria in detail.

-Changes were made to the methods and abstract.

2.The conclusion of ABSTRACT is to clarify what the findings suggest and the theoretical value shown, rather than an outlook for future research.

-Changes were made to the conclusion section to show the theoretical value. 

INTRODUCTION

1.Lines 48-52 have no citations, and the order of references in the manuscript is out of order, and is not cited in ascending order. Author checked and revised.

2.SES does not include its full name on its first appearance. There are also MDD, ODD, SUD, etc. in Table 1. In fact, the authors should check the full text and correct it.

Corrected.

3.Lines 76 to 79, the purpose of the study is to determine the association of different genetic/environ mental risk factors, adverse childhood experiences, and comorbid psychiatric disorders with ADHD and its specific clinical presentation. This purpose appears to have covered lines 80-82 "The study also . . . profile of ADHD." I think the author can adjust the sentence expression and integrate it, otherwise it seems repetitive.

Corrected. Repetition removed

METHOD AND RESULTS

1.It is important that the statistical methods of the article should be listed using a separate subheading.

statistical methods of the article listed as 3.4

2.The author describes the results in lines 143 to 145, 220 to 221, and 271 to 272 using a "correlation" statement, how the author's relevant conclusions are obtained, because the author did not use correlation analysis or other methods to analyze? Therefore, in Tables 3, 4, and 5, it is inappropriate to use the title "Correlation of..." because the authors did not do a true correlation analysis. The statement "the increase in the number of the adversity factors increases the severity of ADHD" is acceptable.

Correlation changed to association

3.Figures 1 and 2 are not labeled in the interpretation of the results, although it may be in lines 141 to 145.

Corrected

4.In Table 2, the author performed a 3-group chi-square test, and the p-value was statistically significant. Did the author conduct further pairwise comparisons to test which groups are different?

Pairwise comparisons to test which groups were not conducted.

5.The authors aim to examine the association of genetic/environmental risk factors, adverse childhood experiences, and comorbid psychiatric disorders with ADHD and its specific clinical presentation. It should be clear which factors are genetic risk factors and which are environmental risk factors. In addition, did the authors exclude the influence of genetic factors when exploring the relationship between environmental risk factors and ADHD?

The influence of genetic factors is highly cited in the previous literature.

Some risk factors such as developmental delay have both possible genetic and environmental etiology, so it is difficult to come to a definite conclusion in our study regarding the specific influence of genetic factors. We combined both genetic and environmental risk factors. We added that statement to the discussion with a reference

6.In Table 4, the authors divided the risk factors into ≤6 and ≥7, and then compared each individual risk factor, whether the author excluded the influence of each individual risk factor itself, such as the first History of self -harm in the comparison, did the author exclude the influence of History of self-harm on the results?

It was included in the analysis, but the results were not significant.

Current suicidal ideation, n (%)

0.75271

No

32 (61.5%)

31 (64.6%)

63 (63.0%)

P-Value 

Yes

20 (38.5%)

17 (35.4%)

37 (37.0%)

History of self-harm, n (%)

0.17921

No

16 (30.8%)

21 (43.8%)

37 (37.0%)

Yes

36 (69.2%)

27 (56.3%)

63 (63.0%)

DISCUSSION

1.Lines 232 to 250, the author seems to repeat the results again, the discussion is to discuss the results of the manuscript, rather than simply repeating the results.

Results were removed from the discussion section.

2.In lines 271 to 277, the author says that a relationship between education level and risk factors has been found. The author's discussion is superficial. Is there any previous literature supporting this result? The author should explain.

More discussion on education level and ADHD is provided. Previous literature supporting the results were added.

CONCLUSION

I don't understand the author's conclusion. It seems that it is all about the prospect and design of future research. The author did not summarize the conclusions of this study? I think this is confusing and the author should rewrite the conclusion section.

The conclusion is rewritten

Reviewer 2 Report

Comments and Suggestions for Authors

Manuscript ID: brainsci-1790666-peer-review-v1

The present study entitled ‘Developmental Risk Factors, Adversity experiences and ADHD Clinical Profiles. A Naturalistic Exploratory Study’ aimed to examine the association of genetic/ environmental risk factors, adverse childhood experiences, and comorbid psychiatric disorders with ADHD and its specific clinical presentation. Authors also aimed to investigate whether the increase in adversity and risk factors increases the severity of ADHD and the number of comorbid psychiatric disorders. For this purpose, they conducted a naturalistic retrospective chart review exploratory study in 100 patients of 16 years or older who have a confirmed diagnosis of ADHD and investigated the presence of developmental risk factors, symptom severity, and comorbidities. Results from qualitative and quantitative analyses showed that the average number of comorbidities increases with the severity of ADHD, like did the average number of risk factors increases; moreover, the number of risk factors positively correlates with the number of comorbidities, but level of education negatively correlates with the number of risk factors and the number of comorbidities.

In general, I think the idea of this study is really interesting and the authors’ fascinating observations on this timely topic may be of interest to the readers of Brain Sciences. However, some comments, as well as some crucial evidence that should be included to support the author’s argumentation, needed to be addressed to improve the quality of the manuscript, its adequacy, and its readability, in particular reshaping parts of the Introduction and Methods sections by adding more evidence and theoretical constructs.

Please consider the following comments:

·         Abstract: According to the Journal’s guidelines, the abstract should be a total of about 200 words maximum and should be introduced as a single paragraph, following the style of structured abstracts, but without headings. Please correct the actual one.

·         In general, I recommend authors to use more evidence to back their claims, especially in the Introduction of the study, which I believe is currently lacking. Thus, I recommend the authors to attempt to deepen the subject of their manuscript, as the bibliography is too concise: nonetheless, in my opinion, less than 50 articles for a literature review are really insufficient. Indeed, currently authors cite only 29 papers, and they are too low. Therefore, I suggest the authors to focus their efforts on researching more relevant literature: I believe that adding more studies and reviews will help them to provide better and more accurate background to this study.

·         Introduction: The authors took a narrow view of mechanisms of association of different genetic/environmental risk factors, adverse childhood experiences, and comorbid psychiatric disorders with ADHD, focusing on how risk factors might increase the severity of ADHD and the number of comorbid psychiatric disorders. Still, I believe that more information about pathophysiology, more specifically about frontolimbic dysfunction involved in altered inhibitory control associated with this disorder would provide a better and more accurate background. Thus, I suggest the authors to make such effort to provide a brief overview of the pertinent published on brain dysfunctions in ADHD, as this information is not highlighted in the text. In this regard, I believe focusing on links between frontal abnormalities and behavioral features of disorders: recently, a novel manuscript provided an overview of the anatomical–functional interplay between the prefrontal cortex and heart-related dynamics in human emotional conditioning (learning) and proposes a theoretical model to conceptualize these psychophysiological processes, the neurovisceral integration model of fear (NVI-f) that can be impaired in the context of psychiatric and neurodevelopmental disorders (https://doi.org/10.1016/j.tins.2022.04.003), while another recent study demonstrated, on a neurophysiological level, the role of PFC in fear conditioning (https://doi.org/10.1111/psyp.14122). Secondary, authors may also consider adding evidence that target pathomechanisms of neurodevelopmental disorders, searching for novel targets, and developing new neuroprotective agents against these psychiatric diseases (https://doi.org/10.1007/s00702-022-02513-5; https://doi.org/10.3390/biomedicines9070734). Finally, authors may also check some additional studies that focused on this topic (https://doi.org/10.1186/s12916-020-1495-2; https://doi.org/10.3389/fpsyt.2019.00673).

·         Data Collection and Analysis: Could the authors provide more detail about the criteria and diagnostic tests utilized to assess comorbid disorders in ADHD patients?

·         Results: I have few concerns about age range of participants, which is too wide. This may reduce the power of the study, therefore I suggest to report a power analysis that will determine the sample size that is most suitable to gain level of significance.

·         In my opinion, I think the ‘Conclusions’ paragraph would benefit from some thoughtful as well as in-depth considerations by the authors, because as it stands, it is very descriptive but not enough theoretical as a discussion should be. Authors should make an effort, trying to explain the theoretical implication as well as the translational application of their research.

·         In according to the previous comment, I would ask the authors to include a ‘Limitations and future directions’ section before the end of the manuscript, in which authors can describe in detail and report all the technical issues brought to the surface. Consider, among the limitations, that proactive inhibitory control has not been assessed.

·         Figures: Please change the scale of the vertical axis and use the same minimum/maximum scale value in all the graphs.

Overall, the manuscript contains 6 tables, 3 figures and 29 references. The number of references is too low for an original research article, and this prevents the possibility of publishing it in this form – in my opinion. However, the manuscript might carry important value describing how an higher number of adversity risk factors correlates with a severe clinical profile in ADHD as manifested by the number of ADHD symptoms and the number of comorbid disorders in patients.

Best regards,

Reviewer

Author Response

Thanks so much for the very helpful comments. The suggestions are excellent and we incorporated all of them. 

Thanks again. 

Dear Reviewer II

Thanks so much for the time and effort you spent reviewing the article. It is highly appreciated, and your suggestions are extremely valuable. We followed all your recommendations.

Please see the specific response to each item.  We also highlighted the changes in blue.

The authors

Reviewer II

Manuscript ID: brainsci-1790666-peer-review-v1

The present study entitled ‘Developmental Risk Factors, Adversity experiences and ADHD Clinical Profiles. A Naturalistic Exploratory Study’ aimed to examine the association of genetic/ environmental risk factors, adverse childhood experiences, and comorbid psychiatric disorders with ADHD and its specific clinical presentation. Authors also aimed to investigate whether the increase in adversity and risk factors increases the severity of ADHD and the number of comorbid psychiatric disorders. For this purpose, they conducted a naturalistic retrospective chart review exploratory study in 100 patients of 16 years or older who have a confirmed diagnosis of ADHD and investigated the presence of developmental risk factors, symptom severity, and comorbidities. Results from qualitative and quantitative analyses showed that the average number of comorbidities increases with the severity of ADHD, like did the average number of risk factors increases; moreover, the number of risk factors positively correlates with the number of comorbidities, but level of education negatively correlates with the number of risk factors and the number of comorbidities.

In general, I think the idea of this study is really interesting and the authors’ fascinating observations on this timely topic may be of interest to the readers of Brain Sciences. However, some comments, as well as some crucial evidence that should be included to support the author’s argumentation, needed to be addressed to improve the quality of the manuscript, its adequacy, and its readability, in particular reshaping parts of the Introduction and Methods sections by adding more evidence and theoretical constructs.

Please consider the following comments:

  • Abstract: According to the Journal’s guidelines, the abstract should be a total of about 200 words maximum and should be introduced as a single paragraph, following the style of structured abstracts, but without headings. Please correct the actual one.

      The abstract is now a single paragraph with a total of 212 words. 

  • In general, I recommend authors to use more evidence to back their claims, especially in the Introduction of the study, which I believe is currently lacking. Thus, I recommend the authors to attempt to deepen the subject of their manuscript, as the bibliography is too concise: nonetheless, in my opinion, less than 50 articles for a literature review are really insufficient. Indeed, currently authors cite only 29 papers, and they are too low. Therefore, I suggest the authors to focus their efforts on researching more relevant literature: I believe that adding more studies and reviews will help them to provide better and more accurate background to this study.

      More reference included

  • Introduction: The authors took a narrow view of mechanisms of association of different genetic/environmental risk factors, adverse childhood experiences, and comorbid psychiatric disorders with ADHD, focusing on how risk factors might increase the severity of ADHD and the number of comorbid psychiatric disorders. Still, I believe that more information about pathophysiology, more specifically about frontolimbic dysfunction involved in altered inhibitory control associated with this disorder would provide a better and more accurate background. Thus, I suggest the authors to make such effort to provide a brief overview of the pertinent published on brain dysfunctions in ADHD, as this information is not highlighted in the text. In this regard, I believe focusing on links between frontal abnormalities and behavioral features of disorders: recently, a novel manuscript provided an overview of the anatomical–functional interplay between the prefrontal cortex and heart-related dynamics in human emotional conditioning (learning) and proposes a theoretical model to conceptualize these psychophysiological processes, the neurovisceral integration model of fear (NVI-f) that can be impaired in the context of psychiatric and neurodevelopmental disorders (https://doi.org/10.1016/j.tins.2022.04.003), while another recent study demonstrated, on a neurophysiological level, the role of PFC in fear conditioning (https://doi.org/10.1111/psyp.14122). Secondary, authors may also consider adding evidence that target pathomechanisms of neurodevelopmental disorders, searching for novel targets, and developing new neuroprotective agents against these psychiatric diseases (https://doi.org/10.1007/s00702-022-02513-5; https://doi.org/10.3390/biomedicines9070734). Finally, authors may also check some additional studies that focused on this topic (https://doi.org/10.1186/s12916-020-1495-2; https://doi.org/10.3389/fpsyt.2019.00673).

      The suggested references were very helpful and they were incorporated in our study.

  • Data Collection and Analysis: Could the authors provide more detail about the criteria and diagnostic tests utilized to assess comorbid disorders in ADHD patients?

      Done. All patients had a full psychiatric assessment by a psychiatrist that included all the possible comorbidities.

  • Results: I have few concerns about age range of participants, which is too wide. This may reduce the power of the study, therefore I suggest reporting a power analysis that will determine the sample size that is most suitable to gain level of significance.

      Power analysis was not conducted since this study was an exploratory study.

  • In my opinion, I think the ‘Conclusions’ paragraph would benefit from some thoughtful as well as in-depth considerations by the authors, because as it stands, it is very descriptive but not enough theoretical as a discussion should be. Authors should make an effort, trying to explain the theoretical implication as well as the translational application of their research.

    New conclusion section is written. 

  •  

 In according to the previous comment, I would ask the authors to include a ‘Limitations and future directions’ section before the end of the manuscript, in which authors can describe in detail and report all the technical issues brought to the surface. Consider, among the limitations, that proactive inhibitory control has not been assessed.

      Done

  • Figures: Please change the scale of the vertical axis and use the same minimum/maximum scale value in all the graphs.

Overall, the manuscript contains 6 tables, 3 figures and 29 references. The number of references is too low for an original research article, and this prevents the possibility of publishing it in this form – in my opinion. However, the manuscript might carry important value describing how an higher number of adversity risk factors correlates with a severe clinical profile in ADHD as manifested by the number of ADHD symptoms and the number of comorbid disorders in patients.

Now we have 42 references.

Round 2

Reviewer 1 Report

Comments and Suggestions for Authors

I still have doubts about the authors' results section "the number of risk and adversity factors is positively correlated with the number of comorbidities, 4) Level of education is negatively correlated with the number of risk and adversity factors and the number of comorbidities.", Although I have raised this question earlier, the author revised the statement to read "the number of risk and adversity factors is positively associated with the number of comorbidities (figure 2) 4) Level of education is negatively associated with the number of risk and adversity factors and the number of comorbidities.". The author just changed "correlated" to "associated", there seems to be no essential difference between the two words and did not solve my original question.

The author does not have objective indicators to show that the two are related, only subjective conclusions. I am more looking forward to the objective indicators of correlation analysis or other statistical methods, and I would like the author to answer my questions or make revisions.

Author Response

Dear Reviewer

We are very appreciative of your helpful feedback. The results of the correlation analysis below. Will incorporate into the results. 

Below is the Pearson results: 

the number of risk and adversity factors is positively correlated with the number of comorbidities

  • R = 0.858 (strong positive correlation)

Level of education is negatively correlated with the number of risk and adversity factors and the number of comorbidities

Level of education vs average risk factors (5 levels of education: <Gr.12, Gr.12, college, univ, post. G.)

  • R = -0.8911 (strong negative correlation)

Level of education vs average no. comorbidities (5 levels of education: <Gr.12, Gr.12, college, univ, post. G.)

  • R = -0.5808 (moderate negative correlation)

Reviewer 2 Report

Comments and Suggestions for Authors

I thanks the author for their effort to improve the manuscript. I have no further questions.

Author Response

We are extremely grateful for the time and expertise of the reviewer. 

Thanks again. 

The authors